# African Swine Fever in Cameroon: A Review

**DOI:** 10.3390/pathogens10040421

**Published:** 2021-04-01

**Authors:** Ebanja Joseph Ebwanga, Stephen Mbigha Ghogomu, Jan Paeshuyse

**Affiliations:** 1Laboratory of Host-Pathogen Interaction in Livestock, Department of Biosystems, Division of Animal and Human Health Engineering, KU Leuven University, 3000 Leuven, Belgium; ebwanga.ebanjajoseph@kuleuven.be; 2Department of Biochemistry and Molecular Biology, Faculty of Science, University of Buea, Buea P.O. Box 63, Cameroon; stephen.ghogomu@ubuea.cm

**Keywords:** porcine disease, African Swine Fever, genotype, production systems, research gaps

## Abstract

African swine fever (ASF) is a hemorrhagic contagious porcine disease caused by the African swine fever virus. The disease poses enormous problems to the pork industry with pig mortality ranging from 30% to 100%, depending on the virulence of the virus circulating. Cameroon, situated in Central Africa is one of the countries in which the African swine fever virus (ASFV) has been endemic since its first outbreak in 1982. The disease is a major problem to the pig industry causing huge economic losses. A clear and concise review on ASF in Cameroon relating to the entry and current genotype of the virus, epidemiology, pathogenesis and economic impact is lacking. A thorough literature search revealed: (1) The virus entered the country in 1982 and caused the death of 80% of the pigs. (2) All isolates belong to serogroup I and only Genotype I is circulating in Cameroon principally in the domestic cycle as there are neither soft ticks nor warthog in the pig production regions sampled. (3) 70% of the pig farmers are involved in the traditional system of production with local and hybrid breeds of pigs with minimal input. (4) The country is endemic to the virus with huge economic losses. (5) So far, very little research has been effected on ASFV in Cameroon. This review gives a detailed overview of the situation of African swine fever virus (ASFV) in the country along with potential avenues for future research into ASFV in Cameroon.

## 1. Introduction

Cameroon is a low-income country that has a population of 25.8 million inhabitants as of 2019 [1] of which 43.5% live below the national poverty line (income less than 3.1 US dollars/day/capita) [2]. An estimated 24% of its population lives below the international poverty line (income of 1.9 dollars/day/capita). The agricultural sector contributes 20% to the Gross Domestic Product (GDP) with the livestock sector contributing 13% and employing 30% of the rural population. One-third of the households are involved in livestock production of which 23.3% are pig farmers. Livestock rearing provides these families with income and nutrition [2]. The current pig production is estimated to be 2.02 kg/person, which is substantially lower compared to the expected 5 kg/person [3,4]. Cameroon is the highest pork producer in Central Africa with over 3.2 million pigs as of 201 [5,6]. The country does not reach its maximum capacity for reasons such as the fact that the sector is confronted with numerous livestock diseases (e.g., classical swine fever, erysipelas, porcine encephalomyelitis, pasteurellosis, salmonellosis, ASF etc.) of which ASF is of utmost importance. Indeed ASF is a devastating viral hemorrhagic porcine disease that can kill up to 100% of affected domestic pigs [7]. African swine fever virus (ASFV), the causative agent of the ASF is a large, enveloped, double-stranded (ds) DNA virus with a genome ranging from 170 to 190 kbp [8]. Original studies on the Spanish virulent Badajoz 1971 (BA71V) isolate reveal that the virus has 37 A+T rich terminal hairpin loops, terminal crosslinks of 27–35 bases complementary relative to sequence on the opposite end, left and right variable regions and a central conserved region [9]. The central conserved region is 125 kbp long while the left variable region ranges from 28 to 47 kbp and the right variable region being 13 to 36 kbp long. These variable regions are home to tandem arrays of multigene families (MGFs) with different copy numbers resulting from gene deletion, duplication or homologous recombination events, and giving rise to differences in length of the ASFV genomes [9,10]. MGFs 100, 110, 360, and 505/560 are found within the left variable region while MGFs 100, 360 and 505 are found within the right variable region [9,10]. The genome contains 151 to 167 ORFs encoding 54 structural proteins and 100 polyproteins in infected cells [11].

Despite the variation in genome size, ASF viruses have a similar genome structure and replication strategy. ASFV predominantly replicates in perinuclear sites in the cytoplasm of infected cells, which are referred to as viral factories. However, some early-stage replication has also been reported in the nucleus of infected cells, which rapidly declines 6 h post-infection [12]. 

The virus was first classified as an *iridovirus* based on its virion morphology, even though it has terminal inverted repeats, terminal crosslinks, a conserved central region with variable regions at the end of the genome and it exhibits temporal gene expressions as can be seen in certain large dsDNA viruses, such as poxviruses [9]. The reclassification of ASFV as the only member of the *Asfarvirdae* family (*asfar*, ASFV and related viruses) became evident as new knowledge on the molecular biology of the virus surfaced. It is the only known DNA arbovirus that replicates in soft ticks of the genus *Ornithdoros* [9,13].

The warthog and the bush-pig are asymptomatic to the infection while the domestic pig is symptomatic [14]. Studies have shown that the virus has developed mechanisms to evade the host immune response and thus proliferate and survive in the host cells for a long time, even after recovery of the host from the infection [15]. Many viral proteins play important roles in this effect: the viral proteins from A276R, A528R, and I329L do inhibit the induction, impact and blocking of interferon-induced effector proteins [16], A179L induces apoptosis and A224L inhibits the cellular inhibitor of apoptosis, while EP153R increases apoptosis upon viral infection of cells [17]. The DP71L proteins promote proteins synthesis in the cells by recruiting host protein phosphatase to dephosphorylate eLL2α while EP402R and A238L inhibit NF-αB [17].

The first ASF outbreak occurred in 1982 and affected over 80% of the 1.6 million pigs in Cameroon [18], The actual source of entry of the virus into the territory has not yet been identified [18,19]. Since 1982, there have been numerous outbreaks, with the most recent one in 2017 that affected the South West and Northern parts of the country.

Based on the sequencing of the major viral capsid protein p72, 24 genotypes of ASFV have been identified [20,21]. Capsid proteins are used in the genotyping of viruses because they appear to be variable and are thus phylogenetically very informative. Sequencing of the gene, coupled with polymerase chain reaction (PCR), provides results within 48 h and is very rapid and cost-effective, as opposed to restriction fragment length polymorphism (RFLP), which warrants virus extraction with results after 5 to 6 days [20]. It is worth noting that the use of viral p72 for the genotyping of ASFV is mainly for epidemiological and molecular studies as it offers no insight into the virulence of the isolates [20].

This review puts together the available literature on the situation of ASF in Cameroon from the first outbreak through the work done towards the search for a vaccine or an effective tool to combat the disease in the country.

## 2. Nucleotide Sequence Diversity of African Swine Fever Virus Isolates from Cameroon 

The ASF genome is known for the variability in its length, based on gene deletion or duplication in relation to the multigene families [20]. 

Insight into the variability of four different isolates from Cameroon, namely CAM/82, CAM/85, CAM/86, CAM/87 and CAM88, using restrictions enzyme analysis followed by Southern blotting using DNA clones, failed to differentiate between CAM82, CAM85, CAM87 and CAM8. However, this method could differentiate the previously mentioned isolates from CAM86, which differs from the others in a 100 bp fragment from the right terminus, as well as within the central region between 89–91 kbp, when analyzed with Asp718W and Asp718H restriction enzymes respectively. An in-depth look into the restriction enzyme analysis revealed that CAM/87 differs by 100 bp from CAM/82, CAM85 and CAM/88 when digested using Asp718 and Asp718N restriction enzymes, while CAM/82, CAM85 and CAM/88 were indistinguishable. This 100 bp difference occurred in the right terminus. A 400 bp difference was seen between CAM86 and the other isolates when digested with BamHI, EcoRI and Asp718H, while HindIII and NcoI digestion revealed 100 bp and 300 bp differences within the variable region 1 (164–167.5 kbp) and variable region 2 (169.5–172 kbp). The length of the genome of the CAM82 isolate was determined to be 172.9 kbp, 171.0 kbp, 171.4 kbp and 170.5 kbp when digested with BamHI, Asp718, XbaI and EcoRI, respectively, resulting in an average length of 172.4 kbp [22]. Data revealed that CAM/82, CAM/85, and CAM/87 were all isolated within 200 km from the North West and West Regions, whereas CAM 86 was isolated in the city of Limbe in the Southwest Region, 265 km from the border of the West Region [22]. 

The small sequence differences noticed within the genomes of the CAM/82, CAM/85 and CAM/87 isolates were not enough to group them into different groups as their restriction enzyme mapping results were much more similar to each other [22]. CAM/82, CAM/85 and CAM/87 were placed in group I. CAM86 was placed in sub-group 2a because of the additional differences in its genome, despite equality in size with other members of the group. Classification by Wesley and Tuthill. (1984), based on restriction enzymes analysis, placed the isolates from Cameroon into sub-group 3 of group 4 [23]. The classification placed isolates from warthog with similar gel patterns in group I, isolates from domestic pigs with different gel patterns into groups II and III and isolates from different geographical origins having similar gel patterns with KpnI and EcoRI restriction enzymes into group VI [23], whereas the first classification by Ekue et al. (2000) compared the gel patterns of the isolates, irrespective of their geographical origins [22,24].

The further classification of Cameroon isolates into genotype I based on genotyping of the B646L gene, was first illustrated by Bastos et al. (2003), after a long silence in ASFV research in Cameroon [20]. Wade et al. (2019) confirmed the circulating genotype (genotype I) following mass sampling and genotyping of ASFV in Cameroon (see Table 1). 

This analysis suggests that the genotype circulating in Cameroon has remained unchanged for all these years [21]. The sequence analysis of the tetrameric tandem repeats within the B602L gene, performed by Giammarioli et al. (2017), reported the presence of variant A with a 23 tandem repeat sequence of amino acids [25]. Further analysis of isolates between 2010 to 2018 by Wade et al. (2019), reported the presence of three variants with 19, 20, and 21 repeats, corresponding to variants A, B, and C, respectively [21]. Initial hypothesis stated that variant A with 19 repeats, which was present during the 1982 and 2010 outbreak in the North, mutated to variant B, and subsequently to variant C by the addition of an A to the CAST sequence of amino acids [21]. On the other hand, Wade et al. reported the presence of these variants in Cameroon irrespective of the location, which is probably due to the movement of farmers from the Southern to the Northern part of Cameroon to establish farms, or the movement of butchers from one location of the country to another to buy and transport pigs for a ready market in the Southern part of the country [21,26]. As shown by Wade et al., variant A and B seem to be the most predominant variants, as variant C was mostly detected in the Far North region. Most interestingly, the variants circulating in the Southwest and Northwest were not highlighted, which could imply that they never had samples from these regions [21]. Seemingly, the same variants are present in some neighbouring countries. Luka et al. (2015), showed that variant C was detected in Tchad and the Central African Republic, while variant B was detected in Nigeria, suggesting that cross border trade between these countries might have led to the entry of variants B and C into Cameroon [27].

Nigeria presently has five variants with 15, 17a, 17b, 20 and 48 amino acids as opposed to Cameroon where only one of those variants with 20 amino acids is present. This makes it difficult to draw a conclusion regarding the entry of the virus from neighbouring Nigeria because, though this country harbours genotype I, the other variants would have been found in Cameroon as well. New studies will are required to clarify the matter.

## 3. Antigenic Variation 

Two different approaches have been used to study the antigenic status of the Cameroon isolates. The first study, based on the presence of neutralizing antibodies, did not help differentiate the isolates, as either the inoculated animals died too early or there was a lack of in vitro neutralization in the cases that the antisera could be obtained from cell passage attenuated isolates [22]. This is evident since no neutralizing antibodies have been proven to be associated with ASF [22,28,29]. 

The second study, based on hemadsorption inhibition (HAI), was very useful in determining the antigenic status of the Cameroon isolates [22]. Interestingly, isolates within the same serogroup do cross protect against one another, as opposed to isolates in different serogroups [30,31]. The results revealed that the Cameroonian isolates were antigenically similar and belong to serogroup I, along with those from Namibia/86 and the European isolates used in the study, while isolates from Eastern and Southern African were antigenically different from those from Cameroon [22]. The most probable reason for the antigenic difference may be the presence of the soft ticks and warthogs (both involved in the sylvatic cycle) in Eastern and Southern African, coinciding with the presence of more than one genotype per country, as opposed to Cameroon in which the soft tick is absent and only genotype I is circulating [20,32].

## 4. The Epidemiology of African Swine Fever in Cameroon

Cameroon is one of the countries in Central African in which ASF is resurging every year. It took just two months for the disease to spread to all the major pig production regions during the April 1982 outbreak. Research has shown that the disease is known to be permanently present and maintained in the wild suids with the help of the soft ticks (sylvatic cycle) in Eastern and South-Eastern parts of Africa. The soft ticks of the *Ornithodoros* genus are known to be involved in the transmission cycle from wild to domestic pigs and thus partly establishing the domestic cycle, in which transmission is mostly from pig to pig, aggravated by the involvement of human activities [25,27,33]. 

A study conducted by Ekue et al. (1990) in Cameroon, geared towards the search for soft ticks and the presence of the warthog (*Phacochoerus africanus*), showed that the major pig production regions (North West, South West, Littoral, and West Regions) had neither warthog nor soft ticks present. In 2015, the Ministry of Livestock also reported the absence of soft ticks within the pig productions areas in the Northern parts of the country [5,34]. On the other hand, the presence of the hard ticks of the family *Ixodidae* has been reported They are responsible for most of the tick-borne diseases plaguing the livestock industry in Cameroon [35,36]. The veterinary personnel also reported the absence of the soft ticks within the four major pig production regions (North West, South West, Littoral, and West Regions) after having been shown samples obtained from the Pirbright Institute [35].

The absence of soft ticks and warthogs in these regions suggests that the only means of transmission of the virus is either by direct contact between infected and non-infected pigs or indirectly through pork and human activities within the pig industry [18]. In addition, the non-respect of adequate biosecurity measures, the fact that many farms are located within a small area and the incautious attitude of other middlemen involved in the pig industry, greatly contribute to the spread of the virus within the national territory [3,5,18].

A serological investigation conducted between November 1984 and March 1985, following the 1982 outbreak, showed an overall 22.5% (25/111 samples) seropositive in the Northwest region, 9.1% (1/11 samples) in the Southwest region and 17.6% (25/142 samples) in the West region. A second survey from January to June 1988 showed a seropositiveness of 10.1% (54/533 samples) in the West region [22]. Results from outbreak epidemiological studies carried out by the Directorate of veterinary service at the Ministry of Livestock, Fisheries and Animal Industries (MINEPIA) (2012, 2014), Wade et al. (2010 to 2018) and Ngu et al. (2019) showed a seropositiveness of 22.8 ± 2.2% in 2012 (nested PCR) and 20.5 ± 2.4% in 2014 (real-time PCR), 42.8% from 2010 to 2018 (PCR) and 15.2% by PCR and 23.8% by real-time PCR in 2019, respectively. The ELISA results were 15.3 ± 1.6% in 2012, 35.6% from 2010 to 2018, and 15.2% in 2019, respectively. In contrast to the research of Ngu et al., which contained an analysis of samples from three regions (Central, South and Southwest), the two other studies had samples from all the ten regions [5,21,37]. In 2013, an incidence of 12% was reported by the Directorate for Veterinary Service of MINEPIA, while the northern regions had an incidence of 0.2% [37].

## 5. Virulent Studies of Cameroon Isolates of ASFV

The clinical signs and pathological lesions associated with the ASFV vary with the virulence of the isolates [38,39]. The isolates are either of low, moderate, or high virulence, with mortality ranging from 30% to 100%. Generally, the disease is characterized by high fever up to 40 °C, anorexia, vomiting, redness of the skin and certain extremities, conjunctivitis, occasionally bloody diarrhoea, increased respiration, and sudden death in certain cases. The pathological lesions that are mostly observed include, amongst others, lung oedema, enlarged and darkened colour of the spleen, petechial haemorrhage in the kidneys and other organs, haemorrhages in the lymph nodes, heart epicardium, and urinary bladder [14,38,39].

Ekue et al. (1989), demonstrated that the pathogenesis of the CAM/82 isolates, responsible for the over 80% deaths in 1982, was quite similar to those of CAM/86 and CAM/87, even though CAM/82 was a low virulent strain as this was a cell passage isolate and not the original isolate [33]. They showed that pigs infected with CAM/82, CAM/86, and CAM/87 all exhibited similar clinical signs, as did animals infected with most of the other isolates in the European, South American, Caribbean, and West African (ESAC-WA) group [40]. Their work proved that a minimum contact time of 2 h and at most 6 h is required for effective transmission of the virus in the case of CAM/82, resulting in a fever of 40 °C on 10–13 days post-infection (dpi). The CAM/82 isolate also produced moderate clinical signs with low mortality, compared to the mortality observed during the original 1982 outbreak, where pyrexia was observed 3–4 dpi, followed by the death of the animals [19], and to those from certain other African isolates [33]. 

That notwithstanding, the clinical signs (lameness, prostration, inappetence, diarrhoea, incoordination) and gross lesions (haemorrhages in the kidney, fluid in the abdominal and thoracic cavities, haemorrhages in the visceral lymph nodes, ventricles of the heart, gall bladder, small and large intestines) produced in all infected pigs were similar for all isolates. However, in the CAM/86 and CAM/88 infected animals, pyrexia was first observed 3–6 dpi and the duration of illness to death was 6–16 days and 7–21 days, respectively. They had higher mortality rates and viremia was undetectable after 30–45 days and 34 days, respectively, which is consistent with what was observed in animals infected with isolates from Brazil and the Dominican Republic, where viremia was undetectable after 35 and 30 days, respectively and similar clinical signs were described [40].

## 6. Production Systems 

About 70% of the pig farmers in Cameroon are engaged in the traditional system of production, where mostly the local breeds of pigs (Bakweri, Bamileke, Mankon Long Nose, hybrids (local and exotic)) are reared with limited input. The animals are left to fend for themselves and can be seen roaming the villages. About 20% of the farmers practise according to the semi-intensive system of pig production. This system is mostly practised by small family setups, where pens are constructed behind homes and on rented land. They usually produce hybrids pigs that result from crosses between the exotic pig breeds. These families usually group themselves into small common initiative groups (CIG) to get loans and subventions from the government to improve their production [2,4,6,41,42]. The intensive system of pig production is practised by 10% of the farmers that also have modern biosecurity measures in place [19,43]. It is therefore evident that pig production in Cameroon is still greatly practised in the informal sector. Cameroonian annual pork production amounts to 30,000 tons, which is not enough to meet the yearly pork demand of 42,000 tons. To meet the increasing demand of its ever-growing population, pork is imported principally from Tchad (42 billion FCFA/year) [4,44]. The fact that the demand for pork is on a rise, coupled with the increasing use of pigs in traditional ceremonies, generated interest in many families to start pig farming [41,42]. Most of these new farms are located in the suburbs, close to the big cities. The odour, noise, poor drainage and limited space for pig farming in these areas compelled the farmers to operate in the rural areas, though some small pig farms, constructed with locally made materials, can still be seen(un-cemented floor, suspended wooden pens or block housing) on marshy areas or dry land [41,42].

Overall, 80% of the pigs in Cameroon are hybrids resulting from crosses between local and exotic breeds (Large white, Duroc, Landrace, Hampshire) or between the exotic breeds [4]. Research has shown that some of these local breeds are less susceptible to ASF infection as compared to the exotic breeds [14,18].

## 7. The Devastating Effect of ASF in Cameroon

Cameroon had a growing pig population from 1959 to 1981 with an estimated 1.6 million pigs within the national territory [19]. Projects, like the Belgian-Cameroon project on pig amelioration in 1980, were instituted to transform the traditional rearing systems to modern pig farming systems [19], but the emergence of ASFV into Cameroon in April 1982 led to the death of over 73,720 pigs on the national territory. The country suffered huge economic losses of 5,233,180$ in 1982, and up to 25,263,600$ in 1987, which led to the Cameroon Government implementing strict sanitary measures in the affected regions (Littoral, North West, South West, Center, and West) to contain the spread of the disease [19]. The numerous outbreaks that have followed since 1982, left the pig industry ramping [33]. The West Region was and still is, the biggest pig producing region in Cameroon, accounting for 63% of the pig population. This equals an amount of 1.6 million animals, of which 54,432 were reported dead by the registered farmers. However, losses of non-registered farmers are not taken into account, since they did not declare their dead [18]. The veterinary personnel were not able to revert the situation while the farmers stopped burying their dead pigs in their misfortune. Carcasses could be seen dumped along roads and stream banks. Interestingly, the local pigs did not suffer the same fate as the exotic breeds, since they were still seen roaming the villages [18,33]. The epidemic led to a drastic shortage of pork, and many people developing a phobia for pork after seeing carcasses everywhere, causing the prices of other meats to increase by 30%. The economic deficit was enormous, as loans could not be repaid due to the great loss on the part of the farmers, some lost up to 36,000$, while some feed store owners had losses of 26,000$. The nation lost 80% of the estimated 1.6 million pigs during the epidemic [18,45]. By 2010, many of the pig farmers travelled to the northern part of the country and established big pig farms with an established market in the southern part of the country. The movement was prompted by the fact the northern part was African swine fever-free. The next big wave of the disease led to more than 100,000 pigs dead in 2011 within the northern region [26]. Many farmers lost 200 to 500 pigs and had to go out of business. The veterinary service at the time blamed the farmers for their careless attitude as they shun simple biosecurity measures [26].

Over 80% of the animals died in the initial outbreak, but the mortality decreased to 40% with the subsequent outbreaks over five years, as reported by many of the farmers. The continuous presence and circulation of these pigs might explain the reduction in mortality, as well as selection pressure imposed by the recovered animals. Of great interest is the fact that piglets from recovered or convalescence sows show reduced viraemia, delayed onset of clinical signs and higher survival rate, which suggest that the antibodies obtained from colostrum play a major role in protection [22]. 

The local breeds of pigs constituted 20% of the pig population during the initial outbreak, of which some died. Some of the restocking farms housed recovered animals, which did serve as a source of re-entry of the virus into the farms because they were seropositive. Most of the farms located far from the city, wherein the workers work and live in the same place, did not suffer the same fate during the outbreak because contact with the other pig farms was greatly restricted [22,46,47].

## 8. Control of African Swine Fever in Cameroon

Following the disastrous nature of the virus and speculative route of entry into the country [18,19], the government instituted stringent measures both in the infected regions and ASF free regions to stop the spread of the virus [19]. On the one hand, the government implemented measures that included, amongst others, the prohibition of the movement of pigs into and out of the infected zones, disinfection of pigsties and all materials related to pig farming, suspension of feed production for three months; tracking and destruction of sick animals, the prohibition of the importation of pork, and the destruction of waste from the airport [19]. On the other hand, the government created a veterinary laboratory in 1983, the National Veterinary Laboratory (LANAVET), because samples were previously sent to Spain and diagnostical results arrived only three months later [19]. The main purpose of LANAVET was to assist in the rapid diagnosis and production of vaccines for veterinary use, thus permitting policymakers to take rapid actions [48].

By 2015, the country was divided into zones, based on the resurgence of the disease following the 2010 national outbreak. The northern regions (Adamawa, North and the Far North) constituted ‘Zone A’ or low-risk zone, as no case of ASF had been declared there following 2010. The central and littoral regions are called the high-risk zones or ‘Zone B1’ while the East, West, North West, South and South-West Regions are regions of moderate risk (‘Zone B2’), as they still occasionally experience a resurgence of ASF [5,49] (Figure 1). Based on the zonal distributions, the Ministry has instituted strict sanitary measures and contingency plans to fight against this deadly porcine disease [5,49]. The three main objectives are: (1) the reduction and maintenance of the incidence at a low rate, (2) the application of strategic measures and techniques within the whole pig farming sector, and (3) the implementation of continuous epidemiological studies on the virus with the national territory. On a general note, measures include the reinforcement of epidemiological surveillance, active control on transportation of pigs from one zone to the other, sensitization and training of different actors within the pig sector, sanitary and serological control of pigs from Tchad at the different sanitary checkpoints and the active involvement of different zootechnical centres in control and surveillance within the different regions, in collaboration with the regional delegations for livestock with the national territory [5,49].

In the year 2000, a joint project of the Food and Agriculture Organization (FAO) and the Cameroon government was launched. The project has so far helped to reduce the incidence of the disease in Cameroon from 11% in 2000 to less than 0.1% in 2014 through the provision of diagnostics tools (PCR and ELISA tests) and implementation of strict sanitary measures [5,6].

## 9. Research Effected in the Context of Cameroon

There is currently neither vaccine nor antiviral drug available against ASFV. Biosecurity measures when properly followed remains the most effective weapon against the virus. Unfortunately, biosecurity measures are not effective in the traditional and small family setups as the farmers visit other farms either for the crossing of a sow, the purchase of pigs or piglets and the fact that they also allow buyers from unknown areas into their farms [18,41]. I other words, there is a need for an effective vaccine or antiviral drug to efficiently combat the ASFV in Cameroon. The road to the discovery of a vaccine or an antiviral drug for ASF in Cameroon is still a long way, even though the virus has been circulating for quite some time. Groundbreaking work to establish the genotype circulating in Cameroon [22], the absence of soft ticks in the regions sampled [34], the influence of cell passage on the genetic stability of the virus [22], the pathogenicity of the isolates [40], the absence of wild suids in the regions sampled and the antigenic relationship between Cameroonian isolates and those of other African countries and Europe has been laid from 1982 to 1988 [22]. This was followed by a long silence characterized by the absence of articles related to research on ASFV in Cameroon from 1990 to 2019. This silence was only broken following the publication of an article on the genotype circulating in Cameroon from 2010 to 2018 [21]. Therefore, there is a need for more research to be carried out to help in the development of an effective control tool for the subsequent eradication of the virus in Cameroon and the world at large.

## 10. Major Gaps Analysis and Future Research into African Swine Fever in Cameroon

The search for a vaccine or antiviral drug against ASF is of high priority. There is still a lot to be done at the level of Cameroon to reach this goal. Many areas for research are still to be exploited and summarized hereafter. 

1. The search for soft ticks in other Regions and the wildlife reserves in the country.

The work effected by Ekue et al. (1990), [34] on the absence of *Ornithodoros* ticks focused on the major pig production regions of the country while the other regions are still to be exploited as many farmers did move to the northern part to establish pig farms. Cameroon is also blessed with many wildlife reserves (Waza, Dja faunal, Ebo, Eda, etc.) and the search for soft ticks in these reserves will ascertain the presence and involvement of the soft ticks in the epidemiology of ASF in Cameroon.

2. The ASFV status of bush pigs and other wild suids present in these different reserves.

Cameroon has a good number of wildlife reserves wherein bush pigs (*Potamochoerus larvatus*) and other wild suids are present. There is a need to isolate and molecularly characterize the virus in these wild suids and also investigate their involvement in the epidemiology of ASFV in Cameroon.

3. Research into vaccine development either attenuated or sub-unit vaccines.

Research so far has shown promising results with homologous protection as opposed to heterologous protection from attenuated vaccine candidates [50]. Cameroon has only one genotype present and generating attenuated vaccine candidates will be helpful as it might lead to a promising vaccine in the future for the field situation of Cameroon. 

4. The search for host genes related to protection in the case of the local and the wild suids.

Many of the local pigs in the 1982 outbreaks did not suffer the same fate as the exotic breeds [15,44]. Thus, the genes related to protection, and the limited rate of replication of the virus in the local pigs are still to be identified. This will be a very promising area of research as potential immunomodulators could be discovered to help protect the exotic breed from the disease. Interferons might be interesting as well [51].

5. The role of carriers or recovered animals in the spread of ASFV in Cameroon.

Many carriers from previous outbreaks are present in many farms and villages. The potential role of these survivors in the epidemiology of the disease is unknown. It’s worth noting that seropositivity does not imply these animals shed the virus [3,52].

6. The evolution rate of the virus.

Cameroon has moved from just having sub-group A to having sub-groups A, B, and C. What the mutation rate of the virus is and if border trade is responsible for the entry of the other sub-groups into the territory are to be investigated.

7. The survival of the virus in feed and the subsequent contribution of feed mills in the spread of the virus.

The purchase of feed in Cameroon is through commercial feed mills, some of which are owned by pig farmers. These mills constitute a potential milieu for the spread of the virus as farmers from different areas do meet here for the purchase of feed (unpublished data). The involvement of these mills in the epidemiology of the disease, the survival of the virus in feed given the harsh climatic conditions, and the biosecurity measures put in place by the pig farmers who are owners of these mills should be looked into.

8. Strategies to better estimate the impact of the disease on the local farmers.

Most of the estimates on the effects of the virus in Cameroon are based on data obtained from the registered farms. However, many of the local backyard piggeries are not registered and most of them do not declare their death (unpublished data). There is a need for effective tools to include the losses of these farmers when estimating the socio-economic effects of the virus in Cameroon.

9. Strategies to track outbreaks especially when they are not reported.

Many local farmers do suffer the devastating effect of the virus in silence. Their losses are not reported and they turn to sell the sick animals to recover some funds. If effective strategies are put in place, these outbreaks will be tracked even when some farmers do not report them.

10. The involvement of butchers in the epidemiology of ASFV in Cameroon. 

Different regions of the country do experience outbreaks at different times. Pork butchers move from one region to the next to buy and transport naive and or sick animals as a result of shortages experience in other regions due to the virus. If these butchers the cause of outbreaks in other regions or if they help in the movement of the sub-groups within the national territory is still to be determined.

11. The antigenic variation of the isolates in Cameroon.

Early work looking into the antigenic variation of the isolates from Cameroon showed that the isolates were antigenically similar [22] but the movement of pigs across the border and the presence of different subgroups within the country necessitates a look into the antigenic variation of the virus circulating after all these years.

12. The transfer of sera from the recovered local or exotic breeds to naïve animals.

Research has established partial protection of animals from ASFV that received sera from recovered animals when challenged with homologous isolates [53]. Whether naive pigs will gain partial or full protection from homologous isolates when given sera from local or recovered exotic breed or wild suids is still to be determined.

13. The spatial pattern of pig farms in Cameroon.

There is currently no concrete information on the spatial pattern of pig farms in Cameroon. Such information will be very helpful as it will provide a quick and effective way to easily control an outbreak of ASF within a region or the national territory.

14. African swine fever prevalence in the local breeds of pigs in Cameroon.

Some of the local breeds of pigs did not suffer the same fate as the exotic breed as the exotic breeds as was reported by Montgomery and Ekue et al. [14,22], but the actual prevalence of the ASFV in these local breeds is still to be determined. These local breeds are mostly free-range animals and do interact with bush pigs and feral pigs.

## 11. Conclusions 

ASFV is a devastating disease ravaging the pig industry in Cameroon and the world at large. Biosecurity measures with proper implementation of contingency plans are most effective in controlling the spread of the virus as there is still no vaccine or antiviral drug available. However, These measures, are difficult to implement fully in Cameroon due to the highly traditional nature of pig production in place. So far, early research has offered great insight into the genotype circulating in Cameroon, the different subgroups, the mode of transmission of the virus, the pathogenesis of the isolates from 1982 to 1988 and their antigenic relationship, and the production systems. However, subjects such as the evolution rate of the virus, the contribution of wild suids in the different reserves and the search for soft ticks in these reserves, are amongst others, avenues for new research to focus on. A vaccine or an effective control tool is of utmost importance in the fight against the virus in Cameroon and the world at large. 

## Figures and Tables

**Figure 1 pathogens-10-00421-f001:**
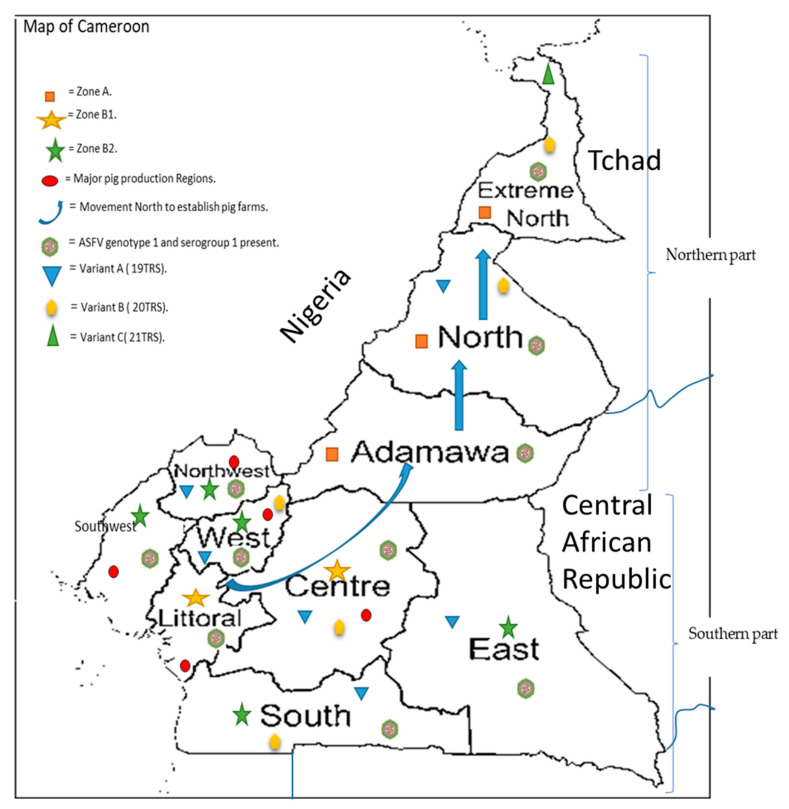
Map of Cameroon indicating the major pig production regions (
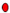
), as well as the different risk zones (zone A (
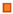
), zone B1 (
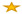
) and zone B2 (
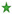
), as partitioned by the Ministry of Livestock after the major outbreak in 2010, the movement of pig farmers (
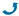
) northwards to establish pig farms for the established market in the southern part. The genotype circulating is genotype I (
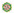
) with all isolates belonging to serogroup 1 and the three variants A (
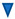
), B (
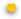
) and C (
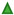
) circulating within the country.

**Table 1 pathogens-10-00421-t001:** Grouping of the ASFV isolates from Cameroon based on different parameters. G1 to G4 represents the major groups while G1a to G4b represents the subgroups within the major groups.

Parameters	G1	G2	G3	G4
G1a	G1b	G2a	G2b	G3a	G3b	G4a	G4b
Similar restriction mapping results with Asp718 restriction enzyme	CAM/82 CAM/85 CAM/87	
An additional difference in sequence when analyzed with Asp718W and Asp718H		CAM/86	
Restriction enzyme analysis (Wesley and Tuthill)		Cameroon isolates	
Genotyping based on the sequencing of the B646L gene	All the Cameroon isolates

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
