# Peer review of "African Swine Fever in Cameroon: A Review"

_pathogens, 2021, doi:10.3390/pathogens10040421_

Round 1

Reviewer 1 Report

The paper is a review on ASF disease in Cameroon including aspects on the socioeconomic impact, epidemiology, genetic characterization, and outbreaks in recent years. The article is publishable but requires an improvement of the English language and rewriting of some parts, as there are repeated the same idea in different sections. There are typographical mistakes that need to be corrected. In my humble opinion, I would reorganize the sections as follows:

- Introduction,

- Nucleotide squence...

- Antigenic variation.

- The epidemiology......

- The pathologenesis of ....

- Production systems

- The devastating effect...

- Control

- Research effected in the context.

- Mayor gaps

Author Response

Reviewer #1:

Q1: The paper is a review on ASF disease in Cameroon including aspects on the socioeconomic impact, epidemiology, genetic characterization, and outbreaks in recent years. The article is publishable but requires an improvement of the English language and rewriting of some parts, as there are repeated the same idea in different sections. There are typographical mistakes that need to be corrected. In my humble opinion, I would reorganize the sections as follows:

- Introduction,

- Nucleotide sequence...

- Antigenic variation.

- The epidemiology......

- The pathogenesis of...

- Production systems

- The devastating effect...

- Control

- Research effected in the context.

- Mayor gaps

A1.1: Per suggestion of reviewers #1 and #3 we rearranged the different sections as to improve the flow for the reader.

A1.2: The inbuilt spelling corrector in Microsoft Word along with Grammarly have been used to spell check the manuscript. Additionally, each sections has been carefully checked and any repetitions were removed as requested by reviewer #1.

Reviewer 2 Report

The revew write by Ebwanga and co-workers is well articolated and conprensible. The topic is acctual and intresting. however I suggest some changed beforore pubblication.

1- Introduction. In line with tilte introduction shold be get start to Cameroom descriptio. So I suggest to start by line 66-79 that add .... Indeed ASF is a devastanting disease that can kill......until line 65.

Than line 79-85.

Moreover, Line 28 insert can before Kill. ASFV can kill about 100% of affected pigs. It depends on virus characteristics.

ASFV is a viral disease that show very important mechanisms of immune escape
that favor the pathogenicity of ASFV.
I think that a paragraph should be insert. E.g. See paper by Reins, Oggiano, Takamatsu, Michael Parkhouse

Author Response

Reviewer #2:

The review write by Ebwanga and co-workers is well articulated and comprehensible. The topic is actual and interesting. however, I suggest some changed before publication.

Q1- Introduction. In line with the title, the introduction should get start to Cameroon description. So I suggest to start by line 66-79 that add .... Indeed ASF is a devastating disease that can kill......until line 65. Then line 79-85. Moreover, Line 28 insert can before Kill. ASFV can kill about 100% of affected pigs. It depends on virus characteristics.

A1: The introduction has been rearranged along with the necessary corrections as requested by reviewer 2.

Q2: ASFV is a viral disease that shows very important mechanisms of immune escape that favour the pathogenicity of ASFV. I think that a paragraph should be inserted. E.g. See paper by Reins, Oggiano, Takamatsu, Michael Parkhouse

A2: A paragraph on ASFV immune evasion has been included in the introduction section as requested by reviewer 2

Reviewer 3 Report

Review: African swine fever in Cameroon by Ebwanga et al.

With the pandemic spread of African swine fever, any information that could help to understand risk factors and their mitigation are needed. Along these lines, the experience of African countries is most valuable, especially when it comes to small scale pig farming in rural areas and under resource-limited conditions.

The authors of the presented review try to summarize the situation in Cameroon and conclude on gaps and ways forward.

While a small review would be valuable beyond doubt (see introductory remarks), the authors do not succeed in providing the data in a concise and well-ordered way. Language and style have much room for improvement (starting with the title: Revew should be Review). Moreover, most figures (especially figures 1 and 2) are not necessary while a figure presenting the geographical situation and the distribution of virus types is completely missing. In my opinion, the manuscript would need major revisions but is of interest.

General points:

  • For a review on ASF in Cameroon, the part on virus characteristics is overlong and there is definitively no need for figure 1 in its current format.
  • Is there a spatial pattern in farm types? (section production systems)
  • Information on pig breeds seems duplicated in the section on production systems
  • With reference to the local breeds and their lower susceptibility: What are the prevalences in these pigs? Were they just not infected due to their behavior or is there higher seroprevalence? If 80% of Cameroon’s pigs died in the first wave of outbreaks, quite some local pigs would have been involved, wouldn’t it?
  • Control section: the information is quite superficial. What measures led to the reduction in disease incidence? Regarding the lab: What took so long in the cooperation with the reference lab in Spain (was it just communication issues)?
  • RFLP pattern should be clearly separated from the sequencing results. What are the genome regions that are linked to the RFLP behavior?
  • Please refer to the summarize of Ekue to clarify the outcome of RFLP analyses as a whole: “Restriction enzyme analysis and restriction enzyme site mapping of genomes of isolates from different parts of the enzootic area failed to distinguish between the 1982, 1985, 1987 and 1988 isolates but the 1986 isolate differed from the others in two fragments occurring within the last lOKbp from the right terminus and another occurring within the central region of the genome (89-91Kbp). The restriction enzyme fragments of the genomes of the Cameroon viruses were similar to the West and South West African isolates from Senegal (Dakar/59), Zaire (Katanga/67) Angola (Angola/70; Angola/72) and Namibia (Namibia/86-1), and were also very similar to the recent European isolates from Malta (1978), Sardinia (1978,1982), Italy (1983), Portugal (1984) and Belgium (1985).”
  • Figure 3 does not clarify the details. Please revise. How do restriction enzyme site maps correlate with the viral genome?
  • I would suggest including antigenic variation into the above section. It should then be called “virus characterization” or something similar.
  • In my opinion, the study published by Wade et al. should be taken into closer consideration. It is mentioned several times, but the full meaning is not presented or set into perspective with RFLP pattern or other findings. Wade et al. (2019) report not only TRS variability (with no correlation in time and space) but also amino acid changes in the partial p72 and changes in the p54. They also report on the tentative spatio-temporal distribution/potential movement and evolution.
  • The same is true for a recent publication by Ngu Ngwa that was apparently not available at the time of manuscript drafting (Ngu Ngwa V, Abouna A, Zoli AP, Attili AR. Epidemiology of African Swine Fever in Piggeries in the Center, South and South-West of Cameroon. Vet Sci. 2020 Sep 1;7(3):123. doi: 10.3390/vetsci7030123. PMID: 32882818; PMCID: PMC7559320). The prevalences and molecular findings presented there should be put into perspective. Is the variant with 19 repeats now prevailing?
  • I would suggest changing “pathogenesis” into virulence studies or biological characterization was mainly clinical and pathomorphological lesions are referred to.
  • How does low virulence of CAM/82 correlate with 80% pig losses in Cameroon?
  • Research section: I recommend to include the information into the other sections.
  • Check if Kouam et al. (2020) needs integration into the section of production systems and/or gap analysis (Kouam MK, Jacouba M, Moussala JO. Management and biosecurity practices on pig farms in the Western Highlands of Cameroon (Central Africa). Vet Med Sci. 2020 Feb;6(1):82-91. doi: 10.1002/vms3.211. Epub 2019 Nov 4. PMID: 31682081; PMCID: PMC7036310.)
  • The gap analysis touches on some important points. However, I suggest adding a gap analysis of the end of each sub-section rather than having findings repeated in the perspective section.
  • The authors refer to many carriers: what is the evidence?
  • What are the anticipated ways to track outbreaks when they are not reported?
  • Do the authors rely anticipate the use of convalescent sera?

Specific and minor comments:

  • Abstract: what is meant by status of the virus?
  • Abstract: points 2 and 4 should be joined as they refer to virus characteristics.
  • Abstract, line 20: correct “endermic”
  • Abstract: avoid primacy claims
  • Introduction: what is the link to vaccine design?
  • Section on epidemiology: The Pirbright Institute is located close to but not in London; refer to UK
  • Figure 2 does not show more than is written in the text; delete
  • Check bibliography, e.g. 20: Journal information; 21: Montgomery, E. (1921)
  • Check spacing, punctuation and comma placement

Author Response

Reviewer #3:

Review: African swine fever in Cameroon by Ebwanga et al.

With the pandemic spread of African swine fever, any information that could help to understand risk factors and their mitigation are needed. Along these lines, the experience of African countries is most valuable, especially when it comes to small scale pig farming in rural areas and under resource-limited conditions.

The authors of the presented review try to summarize the situation in Cameroon and conclude on gaps and ways forward.

Q1. While a small review would be valuable beyond doubt (see introductory remarks), the authors do not succeed in providing the data in a concise and well-ordered way.

A1: We have rearranged the general order as suggested by reviewer 1 to give the review a better flow. The flow is thus:

Introduction,

- Nucleotide sequence...

- Antigenic variation.

- The epidemiology......

- The pathogenesis of...

- Production systems

- The devastating effect...

- Control

- Research effected in the context.

- Mayor gaps

Q2: Language and style have much room for improvement (starting with the title: Revew should be Review).

A2: The inbuilt spelling corrector in word along with Grammarly ( A language software) have been used to spell check the whole document as requested.

Q3: Moreover, most figures (especially figures 1 and 2) are not necessary while a figure presenting the geographical situation and the distribution of virus types is completely missing. In my opinion, the manuscript would need major revisions but is of interest.

A3: Figure 1 will be attached as supplemental for those who may deem it necessary to consult and a map presenting the geographical situation and the distribution of virus types inserted as suggested by the reviewer 3.

General points:

Q4: For a review on ASF in Cameroon, the part on virus characteristics is overlong and there is definitively no need for figure 1 in its current format.

A4: The figure will be added as a supplemental and we wish to keep the virus characterization in its present format since reviewer 1 and 2 found not contrary view to its length.

Q5: Is there a spatial pattern in farm types? (section production systems) add this to the gaps

A5: Information on the spatial pattern in farm types is not available in literature. Hence, we will add this as a gap.

Q6: Information on pig breeds seems duplicated in the section on production systems

A6: This has been revised and the duplication removed.

Q7: With reference to the local breeds and their lower susceptibility: What are the prevalences in these pigs? Were they just not infected due to their behaviour or is there higher seroprevalence?

A7: No information on the prevalence of ASFV in local pigs is available. We have added this as a gap.

Q8: If 80% of Cameroon’s pigs died in the first wave of outbreaks, quite some local pigs would have been involved, wouldn’t it?

A8: As stated by Ekue et al in their paper “Infection of Pigs with the Cameroon Isolate (Cam/82)of African Swine Fever Virus”, some of the local breeds did survive the infection while others died. So the general report had the pigs in Cameroon without emphasising which breed died as opposed to which.

Q9: Control section: the information is quite superficial. What measures led to the reduction in disease incidence? Regarding the lab: What took so long in the cooperation with the reference lab in Spain (was it just communication issues)?

A9: More information (lines 339 to 341 and lines 348 to 364) has been added to the section as requested by reviewer 3 but as to the reason for the delay, there is no information available. In our humble opinion, communication might have been the issues since the rapid communication system in place today was absent then.

Q10: RFLP pattern should be clearly separated from the sequencing results. What are the genome regions that are linked to the RFLP behaviour?

A10.1: The sections have been separated as suggested by reviewer 3.

A10.2: They used different figures to represent the different fragments obtained with respect to the restrictions enzymes which we can’t represent in a single figure. Examples below

Q11: Please refer to the summarize of Ekue to clarify the outcome of RFLP analyses as a whole: “Restriction enzyme analysis and restriction enzyme site mapping of genomes of isolates from different parts of the enzootic area failed to distinguish between 1982, 1985, 1987 and 1988 isolates but the 1986 isolate differed from the others in two fragments occurring within the last lOKbp from the right terminus and another occurring within the central region of the genome (89-91Kbp). The restriction enzyme fragments of the genomes of the Cameroon viruses were similar to the West and South West African isolates from Senegal (Dakar/59), Zaire (Katanga/67) Angola (Angola/70; Angola/72) and Namibia (Namibia/86-1), and were also very similar to the recent European isolates from Malta (1978), Sardinia (1978,1982), Italy (1983), Portugal (1984) and Belgium (1985).”

A11: This outcome has been adjusted ( from lines 96 to 115) as requested by reviewer 3

Q12: Figure 3 does not clarify the details. Please revise. How do restriction enzyme site maps correlate with the viral genome?

A12: Figure 3 has been removed since it cannot represent all the different restriction analysis effected.

Q13: I would suggest including antigenic variation into the above section. It should then be called “virus characterization” or something similar.

A13: We will prefer to have it in a separate section since its gives a clearer picture than being within another section.

Q14: In my opinion, the study published by Wade et al. should be taken into closer consideration. It is mentioned several times, but the full meaning is not presented or set into perspective with RFLP pattern or other findings. Wade et al. (2019) report not only TRS variability (with no correlation in time and space) but also amino acid changes in the partial p72 and changes in the p54. They also report on the tentative Spatio-temporal distribution/potential movement and evolution.

A14: This has been well noted. We made mention of their findings and conclusions in the review (lines 141 to 154).

Q15: The same is true for a recent publication by Ngu Ngwa that was apparently not available at the time of manuscript drafting (Ngu Ngwa V, Abouna A, Zoli AP, Attili AR. Epidemiology of African Swine Fever in Piggeries in the Center, South and South-West of Cameroon. Vet Sci. 2020 Sep 1;7(3):123. doi: 10.3390/vetsci7030123. PMID: 32882818; PMCID: PMC7559320). The prevalences and molecular findings presented there should be put into perspective. Is the variant with 19 repeats now prevailing?

A15: The paper is of great interest and has been added ( lines 222 to 235) as suggested by reviewer 3. We cannot tell if variant A  is the prevailing variant since variant B as well happens to be spreading mostly North.

Q16: I would suggest changing “pathogenesis” into virulence studies or biological characterization was mainly clinical and pathomorphological lesions are referred to.

A16: Adjustment (line 236) made as suggested by reviewer 3.

Q17: How does low virulence of CAM/82 correlate with 80% pig losses in Cameroon?

A17: What they used is their study was not the original isolate but cell culture adapted isolate. Comment adjusted in the manuscript (line 248).

Research section: I recommend to include the information into the other sections.

Q18: Check if Kouam et al. (2020) needs integration into the section of production systems and/or gap analysis (Kouam MK, Jacouba M, Moussala JO. Management and biosecurity practices on pig farms in the Western Highlands of Cameroon (Central Africa). Vet Med Sci. 2020 Feb;6(1):82-91. doi: 10.1002/vms3.211. Epub 2019 Nov 4. PMID: 31682081; PMCID: PMC7036310.)

A18: Work from Kouam et al have been included (lines 282 to 288) as suggested by reviewer 3.

Q19: The gap analysis touches on some important points. However, I suggest adding a gap analysis of the end of each sub-section rather than having findings repeated in the perspective section.

A19: We will prefer to have this as a separate section as it will be more impactful than being in separate sections. We wish it stays as a separate section since reviewer 1 and 2 had no contrary view.

Q20: The authors refer to many carriers: what is the evidence?

A20: Ekue et al in their paper “Infection of Pigs with of African the Cameroon Swine Fever Isolate Virus” wrote that the recovered local pigs served as a source of infection for the others.

Q21: What are the anticipated ways to track outbreaks when they are not reported?

A21: That handicap is one of the problems suggested in the gap analysis.

Q22: Do the authors rely anticipate the use of convalescent sera?

A22: convalescence sera have been proven to offer some protection in literature published elsewhere. The paper titled “African swine fever convalescent sows: subsequent pregnancy and the effect of colostral antibody on challenge inoculation of their pigs”

Specific and minor comments:

Q23: Abstract: what is meant by status of the virus?

A23: By status, we were referring to the genotype present.

Q24: Abstract: points 2 and 4 should be joined as they refer to virus characteristics.

A24: This has been merge as suggested by reviewer 3.

Q25: Abstract, line 20: correct “endermic”

A25: This has been corrected to endemic.

Q26: Abstract: avoid primacy claim

A26: The claim to Primacy has been removed.

Q27: Introduction: what is the link to vaccine design?

A27: The link a vaccine design is  looked at as the research that has been effected in Cameroon so far towards getting a potential vaccine.

Q28: Section on epidemiology: The Pirbright Institute is located close to but not in London; refer to UK

A28: The word London has been removed ( lines 214).

Q29: Figure 2 does not show more than is written in the text; delete

A29: The figure has been deleted.

Q30: Check bibliography, e.g. 20: Journal information; 21: Montgomery, E. (1921)

A30: Bibliography has been checked and all updates effected

Q31: Check spacing, punctuation and comma placement

A31:The inbuilt spelling corrector in word along with Grammarly have been used to spell check the whole document.

Round 2

Reviewer 2 Report

The revision have improved the paper. However the bibliography is a bit old and  should be improved with more recente papers.

I suggest an accurate reading, there are still typos

Author Response

Reviewer #2:

The revision has improved the paper.

  1. We thank the reviewer to provide us with useful suggestion and comments in order to improve our manuscript.

Q1. However, the bibliography is a bit old and should be improved with more recent papers.

A1 As per the suggestion of the reviewer we now updated our bibliography.

Q2. I suggest an accurate reading, there are still typos.

A2. We did a thorough reading of the paper to correct all typos.

Reviewer 3 Report

The article has been improved and I would only suggest minor amendments:

Line 17: „one“ serogroup?

Line 19: check capitalization of „Regions“

Line 40 et seq.: I am still not convinced that details on hairpin structures and tandem repeats are needed in this particular review. For me, nature (dsDNA) and overall size of the genome would suffice. A note on the regions that are routinely sequenced could be added. However, the information does not harm.

Lines 60-63: I do not think that the old taxonomy helps understanding the situation in Cameroon. I would delete this information. However, the above said is true here as well (no harm).

Line 58: Check spacing “6hrs”

Line 234: Virulence rather than virulent

Line 328: Consider including a reference to Schlafer et al. [1,2] when talking about maternal immunity

Line 420: Please state “potential” carriers. Seropositivity does not mean virus shedding (see [3])

  1. Schlafer, D.H.; McVicar, J.W.; Mebus, C.A. African swine fever convalescent sows: Subsequent pregnancy and the effect of colostral antibody on challenge inoculation of their pigs. Am J Vet Res 1984, 45, 1361-1366.
  2. Schlafer, D.H.; Mebus, C.A.; McVicar, J.W. African swine fever in neonatal pigs: Passively acquired protection from colostrum or serum of recovered pigs. Am J Vet Res 1984, 45, 1367-1372.
  3. Stahl, K.; Sternberg-Lewerin, S.; Blome, S.; Viltrop, A.; Penrith, M.L.; Chenais, E. Lack of evidence for long term carriers of african swine fever virus - a systematic review. Virus Res 2019, 272, 197725.

Author Response

Reviewer #3:

The article has been improved and I would only suggest minor amendments:

Q3. Line 17: „one“ serogroup?

A3. This has been corrected to serogroup I

Q4. Line 19: check the capitalization of „Regions“

A4. The capitalizations of regions have all been checked and those still capitalized refer to specific regions of the country.

Q5. Line 40 et seq.: I am still not convinced that details on hairpin structures and tandem repeats are needed in this particular review. For me, nature (dsDNA) and the overall size of the genome would suffice. A note on the regions that are routinely sequenced could be added. However, the information does not harm.

A5. We will prefer to keep the information mentioned by the reviewer in our manuscript. As for the information on the regions that are routinely sequenced, we already added the two most important regions that have been used so far and will wish to preserve this information in the manuscript.(line 138 to line 150)

Q6. Lines 60-63: I do not think that the old taxonomy helps to understand the situation in Cameroon. I would delete this information. However, the above said is true here as well (no harm).

A6. We will prefer to keep the old taxonomy in order to a complete overview for readers that are not acquainted with the ASFV field.

Q7. Line 58: Check spacing “6hrs”

A7. Corrected to 6 hours (line 61)

Q8. Line 234: Virulence rather than virulent

A8. This has been adjusted to virulent (line 255)

Q9. Line 328: Consider including a reference to Schlafer et al. [1,2] when talking about maternal immunity

A9. Reference included (line 409)

Q10. Line 420: Please state “potential” carriers. Seropositivity does not mean virus shedding (see [3])

  1. Schlafer, D.H.; McVicar, J.W.; Mebus, C.A. African swine fever convalescent sows: Subsequent pregnancy and the effect of colostral antibody on challenge inoculation of their pigs. Am J Vet Res 1984, 45, 1361-1366.
  2. Schlafer, D.H.; Mebus, C.A.; McVicar, J.W. African swine fever in neonatal pigs: Passively acquired protection from colostrum or serum of recovered pigs. Am J Vet Res 1984, 45, 1367-1372.
  3. Stahl, K.; Sternberg-Lewerin, S.; Blome, S.; Viltrop, A.; Penrith, M.L.; Chenais, E. Lack of evidence for long term carriers of african swine fever virus - a systematic review. Virus Res 2019, 272, 197725.

A10. All of the above suggestion mentioned in Q10 have been included where appropriate. Paper 1 and 2 (line 409  ) and paper 3 (line 459)